# Injectable DNA Hydrogel-Based Local Drug Delivery and Immunotherapy

**DOI:** 10.3390/gels8070400

**Published:** 2022-06-24

**Authors:** Qi Wang, Yanfei Qu, Ziyi Zhang, Hao Huang, Yufei Xu, Fengyun Shen, Lihua Wang, Lele Sun

**Affiliations:** 1School of Life Sciences, Shanghai University, Shanghai 200444, China; 21722993@shu.edu.cn (Q.W.); quyanfei@shu.edu.cn (Y.Q.); ziyiiiii@shu.edu.cn (Z.Z.); ydssxdxw@shu.edu.cn (H.H.); xuyufei@shu.edu.cn (Y.X.); 2School of Chemistry and Chemical Engineering, Shanghai Jiao Tong University, Shanghai 201240, China; 3Shanghai Advanced Research Institute, Chinese Academy of Sciences, Shanghai 201210, China; wanglihua@sinap.ac.cn

**Keywords:** DNA, hydrogel, drug delivery, immunotherapy

## Abstract

Regulated drug delivery is an important direction in the field of medicine and healthcare research. In recent years, injectable hydrogels with good biocompatibility and biodegradability have attracted extensive attention due to their promising application in controlled drug release. Among them, DNA hydrogel has shown great potentials in local drug delivery and immunotherapy. DNA hydrogel is a three-dimensional network formed by cross-linking of hydrophilic DNA strands with extremely good biocompatibility. Benefiting from the special properties of DNA, including editable sequence and specificity of hybridization reactions, the mechanical properties and functions of DNA hydrogels can be precisely designed according to specific applications. In addition, other functional materials, including peptides, proteins and synthetic organic polymers can be easily integrated with DNA hydrogels, thereby enriching the functions of the hydrogels. In this review, we first summarize the types and synthesis methods of DNA hydrogels, and then review the recent research progress of injectable DNA hydrogels in local drug delivery, especially in immunotherapy. Finally, we discuss the challenges facing DNA hydrogels and future development directions.

## 1. Introduction

Hydrogel is a kind of polymer with three-dimensional network structure formed by physical or chemical cross-linking of macromolecules [1,2,3]. Due to its rich cross-linked network structure and good swellability, in recent years, hydrogels have been widely used as drug carriers in transdermal drug delivery systems to achieve sustainable and controlled drug release [4]. Among them, hydrogels formed by self-assembly of single-stranded DNA and that formed by using DNA as a cross-linking agent have attracted much attention due to the unique properties of DNA molecules [5,6]. DNA is intrinsically the carrier of biological genetic information, and its chemical essence is a hydrophilic linear block copolymer composed of four kinds of deoxynucleotides (A, T, C, G). Natural DNA usually forms a double helix structure by complementary base pairing (A-T, C-G) of two single-stranded DNAs. Since Alexander et al. discovered that single-stranded DNA can spontaneously form DNA double helix in solution [7], and Caruthers et al. invented the automated solid-phase synthesis of DNA [8]. DNA had been tried to construct two-dimensional and three-dimensional objects. Remarkably, in 1996, Nagahara et al. designed the first DNA-based hydrogel via the cross-linking of single-stranded (ss) DNA grafted on polyacrylamide chains [9]. In 2006, Luo et al. constructed the first hydrogel composed of DNA alone. Since then, DNA hydrogels have experienced tremendous development, and been pervasively used in biomedicine and bioanalysis [10,11,12].

At present, DNA hydrogels are mainly divided into two categories, namely the pure DNA hydrogel and hybrid DNA hydrogel [13,14,15,16]. Generally, pure DNA hydrogels formed via the DNA hybridization induced cross-linking of branched DNA monomers with sticky ends. Initially, simple Y- or X-type DNA monomers consisting of 3–4 strands were used to construct DNA hydrogels [17,18,19,20]. Now, more complex DNA monomers such as polypody-type and Takumi-type were also developed, all of which have their own advantages in drug delivery [21,22]. Although the self-assembly of DNA monomers represents a convenient way to construct DNA hydrogels, it requires the consumption of many DNA strands. Therefore, rolling circle amplification (RCA) had also been introduced to reduce the cost of DNA hydrogels, in which a large number of long DNA strands with periodic repeat sequences were produced with the help of enzymes to form hydrogels via physically entangle [23,24]. Hybrid DNA hydrogels were usually constructed through DNA mediated cross-linking of other materials. At present, many biocompatible polymers including polyacrylamide, chitosan and PEG, had been introduced into hydrogels [25,26,27]. In addition, nanoparticles such as liposomes [28], black phosphorus quantum dots [29], carbon dots [30], silica nanoparticles [31] and metal nanoparticles [32] have also been cross-linked with DNA to form hydrogels. These hybrid hydrogels could integrate the functions of DNA and other materials, thus broadening the applications of DNA hydrogels.

Compared with other polymer hydrogels, DNA hydrogels have several advantages. First, since the sequence of artificially synthesized DNA strands can be edited arbitrarily, DNA hydrogels have good programmability. For instance, changing the size and number of branched DNA monomers can easily adjust the thermal stability and viscoelasticity of DNA hydrogels [33], which makes the physical properties of DNA hydrogels more controllable. Second, since DNA is excellent in hydrophilicity, and negatively charged under the pH of body fluids, DNA hydrogels stabilized by non-covalent interactions have superior biocompatibility compared to hydrogels prepared by chemical cross-linking [5,6]. Moreover, due to the presence of DNA-degrading enzymes in organisms, DNA hydrogels can also be degraded into fragments small enough to be absorbed by cells through endocytosis [34], thus having better biodegradability. In addition, DNA can be easily chemically modified with other functional materials during its synthesis, thereby endowing DNA hydrogels with good recognition ability and designable responsiveness that is beneficial for drug delivery [35,36]. DNA hydrogels could also be immunologically active when they contain the sequence of CpG oligonucleotide (an immune adjuvant), which makes DNA hydrogels useful in the field of immunotherapy [34].

In this review, we focus on the recent advances of DNA hydrogels and their applications in local drug delivery and immunotherapy (Figure 1). First, we classified and discussed the preparation methods and functions of various DNA hydrogels by listing some typical cases. Then, the applications of DNA hydrogels in local drug delivery were introduced by taking antitumor drugs, drugs for the treatment of allergic conjunctivitis, drugs for the treatment of nerve damage and antibacterial drugs, as examples. The applications of DNA hydrogels in immunotherapy such as immune checkpoint inhibitors delivery, immune adjuvant delivery, tumor vaccine development and synergistic immuno-photothermal therapy were also discussed. Finally, we discuss the prospects and challenges of DNA hydrogels, and clarify future research directions.

## 2. Classification of DNA Hydrogels

### 2.1. Pure DNA Hydrogel

In 2006, Luo and his colleagues constructed the first pure DNA hydrogel [37]. They designed three DNA monomers with palindromic sticky tails, namely T, Y and X-shaped DNA, and the hybridized DNA molecules were connected to each other by T4 ligase to form a hydrogel. This enzymatically cross-linked DNA hydrogel was stable under physiological conditions [38]. Furthermore, the researchers found that self-assembly of DNA monomers is simpler and faster in preparing DNA hydrogels, and it has become the commonly used method in DNA hydrogel constructing [39]. Further, in order to reduce the cost, rolling circle amplification (RCA) had also been introduced to build DNA hydrogels [40], in which enzyme assisted isothermal nucleic acid amplification was used to produce many long DNA strands with periodic repeat sequences [41]. Next, we introduce the two methods commonly used in DNA hydrogel construction.

#### 2.1.1. Self-Assembled DNA Hydrogels

Liu and colleagues reported the first self-assembled Y-shaped pure DNA hydrogel which consisted of two kinds of building blocks, a Y-shaped DNA scaffold and a double-stranded DNA linker [42]. The Y-shaped scaffold composed of three single-stranded DNA (ssDNA) has three sticky ends, which can hybridize with the DNA linker having two sticky ends to form DNA hydrogel. In addition, X-shaped DNA hydrogels are also very common, and compared with Y-shaped, they have a better degree of cross-linking [43]. However, pure DNA hydrogels assembled with X-shaped or Y-shaped tiles usually require thermal annealing and higher DNA concentrations. In this regard, Wu et al. proposed a dendritic DNA hydrogel assembled under isothermal environmental conditions (Figure 1A) [44]. In this work, the 5′ end of the ingeniously designed linker DNA (L) could complement and pair with one of the four branches of the dendritic DNA scaffold, and a palindromic sequence at the 3′ end could hybridize with another L DNA molecule to form a three-dimensional network of hydrogels. The DNA hydrogels described above are all static structures. Studies have shown that i-motif [45], aptamer [46] or other stimulus-responsive units could endow DNA hydrogels with the ability to specifically respond to certain stimulations. For example, Liu and colleagues used Y-shaped DNA structures and i-motif structures to construct a pH-responsive DNA hydrogel [47]. This Y-shaped scaffold had a rigid double stranded central domain and three i-motif sequences located at the ends. When the pH value decreased, some cytosines were protonated, and C···CH^+^ triple hydrogen bonds would form between protonated (CH^+^) and unprotonated cytosines, resulting in cross-linking of adjacent Y units to form hydrogels. The i-motif could undergo rapid transition within seconds, and the formed DNA hydrogel had good stability. In addition, recently, Hu et al. constructed an acid-resistant and physiological pH-responsive DNA hydrogel using A-motif and i-motif structures [48]. At physiological pH, the DNA hydrogel transitions to a liquid phase due to the dissociation of the A-motif and i-motif into A-rich and C-rich single-stranded sequences, respectively. This hydrogel had good resistance to acidic pH in the stomach and duodenum, thus being advantageous in hydrogel-based insulin delivery.

#### 2.1.2. Rolling Circle Amplification (RCA) DNA Hydrogel

Rolling circle amplification (RCA) is an isothermal DNA amplification method that utilizes a short ssDNA complementary to a circular DNA template as a primer to generate a large number of long DNA strands with periodic repeat sequences with the help of enzymes [49]. These long chains can physically entangle to form hydrogels [38]. As early as 2012, Luo and colleagues creatively constructed DNA hydrogels with the help of polymerase Φ29 (Figure 1B) [50]. The DNA chains was elongated by RCA followed by multi-primed chain amplification (MCA), which further forms the DNA hydrogel by physical entanglement. Interestingly, the hydrogels prepared by this method exhibit unique mechanical properties, appearing as a liquid state in the absence of water, but as a solid state in the presence of water. The involvement of RCA in the formation of hydrogels greatly simplifies the design of DNA building blocks and overcomes the limitation of high cost of long DNA chains. However, the preparation of DNA hydrogels by RCA takes a long time. In this regard, Song et al. optimized the preparation of DNA hydrogels by RCA [51]. They introduced L-shaped “soft-brush” DNA strands into the RCA to build DNA hydrogels in 1 min. In addition, functional fragments can also be introduced into DNA strands produced by RCA amplification to prepare DNA hydrogels with more functions. For example, recently, Jin et al. developed an RCA-based DNA hydrogel that integrated dual-target and multivalent aptamers to achieve targeted drug delivery [52].

### 2.2. Hybrid DNA Hydrogels

#### 2.2.1. Polymer-DNA Hybrid Hydrogels

Acrylamide polymers are the most widely used water-soluble polymers in hydrogel preparation [33,53]. As early as 1996, Nagahara et al. had constructed hybrid DNA hydrogels with acrylamide polymers. In this pioneering work, single-stranded DNA was attached to the side chains of Succinimido-copolymers (poly(N,N-dimethylacrylamide-co-N-acryloyloxysuccinimide)). Then, the succinimido-copolymers were cross-linked by DNA hybridization to form a thermoresponsive hydrogel based on the thermosensitivity of double-stranded DNA [9]. This method of cross-linking polymer molecules to form hydrogels by DNA hybridization was widely used in the construction of various polymer-based hybrid DNA hydrogels [25]. In recent years, researchers had also constructed a variety of polymer-DNA hybrid hydrogels with tunable morphological and mechanical properties. For example, Willner et al. used Ag^+^-stabilized i-motif to cross-link poly-N-isopropylacrylamide copolymerchains (pNIPAM) to construct pH-responsive hybrid DNA hydrogels [54]. Since this i-motif structure was only stable at lower pH (5.2), a reversible gel-to-solution transition could be achieved by adjusting pH value. They had also used two single-stranded DNA that could form G-quadruplexes and two complementary DNA strands to cross-link polyacrylamide to form hybrid hydrogels with tunable mechanical properties. (Figure 1C) [55]. Since K^+^ is crucial for the formation of G-quadruplexes, and crown ether can competitively bind K^+^, the mechanical strength of hydrogels can be adjusted by alternately introducing K^+^ and crown ether. In addition, Zhao et al. also used two-stage cross-linking combined with 3D printing technology to construct a morphology-tunable hydrogel device [56]. In this study, polyacrylamide carrying single-stranded DNA was first chemically cross-linked to form a gel, and then the macroscopic morphology of the hydrogel could be rapidly changed by adding complementary bridging DNA strands.

Besides acrylamide polymers, chitosan is also commonly used to construct hybrid DNA hydrogels. Chitosan is a naturally derived positively charged glycopolymer with good biodegradability. Due to the polyanionic character of DNA, chitosan can be easily integrated with DNA hydrogels through electrostatic interactions to form hybrid hydrogels with tunable physical properties. For example, DNA-CS hybrid gels made by cross-linking pre-annealed DNA hydrogels with chitosan (CS) showed various properties that could be fine-tuned, such as porosity and viscosity [57]. In addition, Ishii-Mizuno et al. found that compared with pure DNA hydrogels, the hybrid hydrogels formed by mixing chitosan and DNA hydrogels containing unmethylated cytosine-phosphate-guanine (CpG) sequences showed better stability, toughness, water retention and injectability [58].

In addition, poly (lactic-co-glycolic acid) (PLGA), an FDA-approved drug excipient for clinical use, and polyethylene glycol (PEG) have also been used in the construction of hybrid DNA hydrogels. For example, Zhu et al. used biodegradable PLGA to pre-encapsulate water-insoluble therapeutic agents and then mixed with Y-shaped and cross-linker DNA monomers to prepare DNA/PLGA hybrid hydrogels (HDNA), which could realize effective local release of water-insoluble therapeutic agents [59]. Chen et al. constructed a core-shell hydrogel microsphere based on PEG and DNA sensing device using a microfluidic device [27]. The DNA sensing device in the outer layer triggers a cascade reaction involving the DNA sensing device in the core region after recognizing tumor-associated miRNA, thereby realizing tumor-specific siRNA delivery.

#### 2.2.2. Nanoparticles-DNA Hybrid Hydrogels

In addition to the above organic polymers, in recent years, some organic or inorganic nanoparticles have also been used to construct hybrid DNA hydrogels, including liposomes [28], quantum dots [29], carbon dots [30], etc. These nanoparticles could further increase the functionality of DNA hydrogels. Liposomes are commonly used delivery vehicles of hydrophobic and hydrophilic small-molecule drugs. Cholesterol can intercalate into the phospholipid bilayer of liposomes through hydrophobic interactions. Based on this, Guo et al. used liposomes to non-covalently cross-link polyacrylamide copolymers carrying cholesterol-modified double-stranded DNA to form hybrid hydrogels [28]. In the presence of restriction endonucleases or temperature changes, the hydrogels exhibit a stimuli-responsive gel-to-sol transition, enabling controlled release of liposomes. Quantum dots (QDs) are semiconductor nanocrystals with unique photophysical properties, tunable photoluminescence quantum yields (PLQY) and photostability. BPQDs had high absorption under visible light irradiation, thus being ideal materials for photothermal therapy (PTT) and photodynamic therapy (PDT). Wen et al. had reported a nanophotothermal DNA gel (PEI@BPQDs-DNA gel) for tumor therapy [29]. In their design, negatively charged black phosphorus quantum dots (BPQDs) were first combined with positively charged polyethyleneimine (PEI) to form PEI@BPQDs. Then, the PEI@BPQDs were mixed with X-shaped DNA monomer to make the PEI@BPQDs-DNA gel via electrostatic interaction and hydrogen bonding. Carbon dots (C-Dots) are nanomaterials with the characteristics of photoluminescence, high biocompatibility and low cost. The fluorescent properties of carbon dots give them great potential for tracking drug delivery. For instance, Das et al. had developed a carbon dot-DNA hybrid hydrogel for tracking the release of drug by fluorescence imaging [30]. In their study, amine-functionalized carbon dots were linked to the 5′-phosphate terminus of cytosine (C)-rich ssDNA via phosphoramidate linkages (Figure 1D). Due to the cytosine (C)-rich ssDNA could form i-motif structures under mildly acidic conditions, the sol-gel transition of this carbon dot-DNA hybrid hydrogel could be achieved by changing the pH value of the solution, thereby realizing pH responsive drug release. The fluorescence of carbon dots could be tracked by imaging to correlate the efficacy and level of drug loading.

## 3. DNA Hydrogel-Based Local Drug Delivery

In the past 5 years, DNA hydrogels have shown great potential in local drug delivery for the treatment of different diseases, including cancer, allergic disease, nerve damage and infectious disease [59,60,61,62].

### 3.1. Local Delivery of Antitumor Drugs with DNA Hydrogels

Chemotherapy is one of the most used methods for cancer treatment [63]. However, the systemic administration of chemotherapy drugs has strong side effects, which can even lead to the death of patients [64]. Meanwhile, it has been demonstrated that local drug delivery based on DNA hydrogel could significantly reduce the side effects of drugs [34]. For example, Li et al. constructed DNA hydrogels for local chemotherapy of tumors by grafting camptothecin (CPT) on a DNA backbone [60]. They assembled DNA grafted with CPT into two Y-shaped structural units, which were joined together to form a drug-containing hydrogel (CPT-DNA-Gel) (Figure 2A). CPT-DNA-Gel shows excellent injectability, thermal sensitivity and nuclease response properties. Due to enzymatic degradation, CPT-DNA-Gel can gradually break down into nano-sized particles that can penetrate into residual tumor tissue and be taken up by tumor cells. Moreover, it shows excellent long-term cytotoxicity against cancer cells and effectively prevents cancer recurrence in vivo, while exhibiting low toxicity to normal tissues.

In addition to delivering chemotherapeutic drugs, DNA hydrogels can locally deliver nucleic acid drugs. RNA interference is also an effective treatment strategy, which requires effective delivery of interfering RNA (siRNA) to tumor cells. However, siRNA is easily degraded by nucleases in the humoral environment [65]. To this end, Lee et al. constructed DNA-RNA hybrid hydrogels for tumor-targeted siRNA delivery via dual enzyme polymerization employing both rolling circle amplification (RCA) and rolling circle transcription (RCT) [66]. In their design, the hybridization of multimeric short hairpin RNAs (shRNAs, the product of RCT) and polyvalent functional DNA aptamers (the product of RCA) mediated the formation of the hydrogel, and in order to facilitate the release of functional small interfering RNA (siRNA)-aptamer complexes (SACs), restriction enzyme reaction sites were edited into the nucleic acid sequences (Figure 2B). The resulting hydrogel was ultrasoft and injectable, and achieved tumor cell-targeted siRNA delivery. In addition, Chen et al. synthesize novel multi-compartment DNA hydrogel particles for smart siRNA delivery to tumor cells [27]. The DNA sequence used to synthesize the hydrogel included two parts: the sensing sequence (D1) and the treatment sequence (D2). D1 could interact with the cancer biomarker miRNA-21 to release specific intermediate sequences which would react with D2 to trigger the release siRNA. This DNA hydrogels particles could selectively induce the death of colon cancer cells, thus being promising in siRNA-based tumor treatment.

Photothermal therapy is also a promising direction of tumor treatment which avoids the use of chemotherapy drugs with strong side effects. Generally, photothermal therapy need intravenous injection of nanomaterials with good photothermal conversion efficiency that could be enriched into tumors for tumor cell killing via converting light energy into heat [67]. However, the enrichment efficiency of nanomaterials at tumor sites is often extremely low, and tissue penetration depth of nanomaterials is limited. To this end, Lee et al. reported a positively charged DNA hydrogel containing the photothermal material black phosphorus quantum dots (BPQD) for tumor photothermal therapy [29]. The positive charge carried by the polymer polyethyleneimine (PEI) enhanced the permeability of the DNA hydrogel, allowing it to easily penetrate and translocate in tumor cells. The photothermal effect of black phosphorus quantum dots (BPQD) could effectively cause the apoptosis of tumor cells. The hydrogel cleared tumors 20 days after treatment and significantly reduced drug resistance and improved overall survival in mice with in situ mammary tumors.

### 3.2. Local Delivery for Anti-Inflammatory Drugs with DNA Hydrogels

The clinical need for the treatment of allergic conjunctivitis (AC) is rapidly increasing. However, current over-the-counter medications have limited efficacy and might lead to adverse side effects (e.g., vasodilatory rebound from topical vasoconstrictors; dry mucous membranes or drowsiness from oral antihistamines) [68]. Moreover, AC-related anti-inflammatory compounds are often difficult to dissolve in water, thus limiting their therapeutic potential. Zhu et al. reported a DNA/poly(lactic-co-glycolic acid) (PLGA) hybrid hydrogel (HDNA) for delivery of the water-insoluble ophthalmic drug dexamethasone (DEX) (Figure 3) [59]. The PLGA pre-encapsulation allows for the loading of water-insoluble therapeutic agents, while the porous structure of the HDNA provides a continuous supply of therapeutic agents, enhancing the accumulation and gradual release of DEX. Studies have shown that the complexation of DEX with HDNA (HDEX) significantly increases its cellular uptake by approximately eight-fold and that the drug exhibits a long retention time of 24 h in ocular cells and tissues. Therefore, the HDNA-based ophthalmic treatment delivery systems enable biocompatible and biodegradable treatment paradigms.

### 3.3. Local Delivery of Drug against Nerve Damage with DNA Hydrogels

Peripheral nerve injury (PNI) is a global disease that often results in motor or sensory nerve dysfunction and traumatic neuroma formation [69]. It has been found that vascular endothelial growth factor (VEGF) should be released early in the peripheral nerve regeneration (PNR) to promote polarized angiogenesis, which can provide nutritional support, direct the migration of Schwann Cells (SC) and guide the direction of axonal growth [70]. In contrast, sustained release of nerve growth factor (NGF) has a better repair effect than burst release. Therefore, Zhang et al. developed DNA hydrogels loaded with vascular endothelial growth factor (VEGF) and nerve growth factor (NGF) for PNI repair [61]. In their design, the NGF was loaded into X-type DNA hydrogels, and then DNA ligase was utilized to ligate VEGF-loaded T-type DNA monomers with X-type DNA hydrogels to obtain XT-type DNA hydrogels (Figure 4). Then, the biphasic release of VEGF and NGF was achieved due to the different degradation rates of X- and T-type DNA monomers. The results showed that the DNA hydrogel could promote the proliferation, migration and myelination of Rat Schwann cells (RSC), while maintaining cell viability, thus having good clinical application prospects. The central nervous system, which consists of the brain and spinal cord, is also vulnerable to injury. Mammals with severe spinal cord injuries are unable to regenerate naturally, and transplanting stem cells to the injury site is a very promising approach, but there is an urgent need to address the issue of permeability of the delivery material [71]. Gui et al. reported a supramolecular DNA hydrogel with extremely high permeability carrying homologous neural stem cells (NSCs) for repairing the spinal cord space in rats [72]. The hydrogel composed entirely of DNA double strands to form a molecular network was more permeable than traditional polymer hydrogels. With this hydrogel, NSCs could be implanted into the lesion where they proliferated and differentiated to form a renascent neural network. This supramolecular DNA hydrogel is promising in localized NSCs delivery due to its good permeability, self-healing ability, and proper mechanical support.

### 3.4. Local Delivery of Antimicrobial Drug with DNA Hydrogels

*Staphylococcus aureus* is the most common pathogen in human septic infections [73]. Ee et al. loaded cationic antimicrobial peptides (AMPs) into polyanionic DNA nanostructures to create hydrogels that release AMPs in response to pathogenic *Staphylococcus aureus* infections [62]. The DNA hydrogel consists of Y-scaffold and L-linker where the AMPs binds to the DNA backbone via electrostatic interactions (Figure 5). This DNA hydrogel is cleaved in the presence of endonuclease secreted by pathogenic *S. aureus* infection and then releases AMPs, which have rapid anti-inflammatory activity against *S. aureus*-infected wounds and rapidly promote wound healing.

## 4. DNA Hydrogel-Based Immunotherapy

Tumor immunotherapies that enhance the anti-tumor immune response in patients have shown great potential in fighting against malignant tumors in recent years [74]. Tumor immunotherapies can be divided into systemic immunotherapies or local immunotherapies, the former include systemic immune adjuvant delivery, adoptive cell therapy and cancer vaccines, while the latter is to locally modulate tumor immunosuppressive microenvironment [75]. However, the current clinical application of tumor immunotherapies also encounters some difficulties. For example, systematic administration of immunotherapy drugs, such as immune adjuvants and cytokines, might cause serious side effects such as cytokine storm. Therefore, DNA hydrogel-mediated local drugs delivery has been also explored in tumor immunotherapies.

### 4.1. Immune Checkpoint Blocking

It has been reported that the abnormal overexpression of immune checkpoints in the tumor microenvironment can inhibit the anti-tumor function of immune cells [76]. For instance, the highly expressed programmed death ligand 1 (PD-L1) on the surface of tumor cells could weaken the activity of cytotoxic T cells by interacting with the programmed death receptor (PD-1) expressed on T cells [77]. At present, some immune checkpoint inhibitors have been approved by the FDA, such as Nivolumab, but their high cost limits their wider clinical application [78]. Fortunately, benefiting from the SELEX (systematic evolution of ligands by exponential enrichment) technology, nucleic acid aptamers targeting immune checkpoints have been screened in vitro. However, efficient delivery of these aptamers to tumors while avoiding degradation caused by nucleases is key to achieve immune checkpoint blocking. To this end, Lee et al. prepared a DNA hydrogel for controlled release of PD-1 aptamers triggered by Cas9/sgRNA [79]. The hydrogel was obtained by rolling-circle amplification technology from DNA sequences containing PD-1 aptamers and sgRNA target sequences. When injected with cas9/sgRNA, PD-1 aptamers could be precisely released via cas9 mediated cleavage, followed by binding with PD-1 on CD8^+^ T cells to block the immune checkpoint, thereby enhancing the vitality of CD8^+^ T cells to kill tumor cells. Meanwhile, the author found that the hydrogel stayed longer than the free aptamer at the injection site, which contributed to the excellent antitumor effect.

The PD-1 aptamer delivered by DNA hydrogel could not only block immune checkpoints, but also enrich T cells in tumors. Recently, Yao and his colleagues developed a DNA hydrogel network by rolling-circle amplification to capture T lymphocytes inside tumors for localized immunotherapy (Figure 6A) [80]. In this strategy, DNA hydrogel network was constructed by partially hybridization of two kinds of DNA sequences containing PD-1 aptamers and CpG oligonucleotides (CpG ODN), respectively. PD-1 aptamers in the hydrogel could not only capture tumor infiltrating T cells, but also block PD-1 on T cells. In addition, cutting sites of a restriction enzyme were edited in the DNA sequences for the responsive release of captured T-cells and CpG ODN. The results showed that the high purity and survival rate of the captured tumor infiltrating T cells, in combination with CpG ODN-induced activation of antigen presenting cells, could elicit potent antitumor efficacy.

### 4.2. Immune Adjuvants Delivery

It has been reported that hydrogel-based sustained release of immune adjuvants could effectively activate antitumor immune responses while avoiding serious side effects [81]. CpG ODNs is a class of synthetic DNA fragments containing unmethylated cytosine phosphate guanine (CpG) dinucleotide, which could activate the innate immune system after binding with Toll-like receptor 9 (TLR9) [82,83]; therefore, it can be easily integrated with DNA hydrogels for tumor immunotherapy. As early as 2011, Nishikawa et al. prepared a DNA hydrogel formed by self-assembly of immunostimulatory CpG ODN to stimulate local antitumor immune responses against mouse colon tumor [34]. In addition to CpG ODNs, guanosine and uridine rich single-stranded RNA (GU-rich RNA) can recognize Toll-like receptor 7 (TLR7) and Toll-like receptor 8 (TLR8) [84]. It can also induce strong immune response, but its application is limited by the instability of RNA. Komura and his colleagues developed a GU-rich DNA or RNA hydrogel (RDgel), which was composed of 6 phosphodiester DNA and a 20-mer phosphorothioate-stabilized GU-rich RNA. The result shows that it could continuously release GU rich RNA and induce effective immune reactions against mouse colon tumor [85].

### 4.3. Tumor Vaccine Delivery

Tumor vaccines can elicit potent tumor specific immune reactions, thus being a promising direction in tumor immunotherapy. During the past few decades, tumor-associated antigens such as MUC1 glycopeptide [86], glycolipids [87] and Globo-H [88] have been used as effective anti-tumor vaccine epitopes [89]. In this regard, preparing vaccines with these tumor-associated antigens can stimulate specific immune response to attack tumor cells. Previously, Nishikawa’s team had demonstrated that the assembly of CpG ODNs into polypodna DNA structures (polypodna) could efficiently deliver CpG ODNs to immune cells [90,91]. It was further proved that the DNA hydrogel containing CpG ODNs was a promising tumor antigen delivery system [81]. Even using ovalbumin (OVA) as a model antigen, it could induce strong anti-tumor immunity with few adverse effects [92]. Based on the above research, Umeki and his colleagues designed a CpG DNA hydrogel incorporated with both the tumor antigens and antigen-presenting cells (APCs) for tumor immunotherapy [93]. The results showed that intra-tumoral injection of the DNA hydrogel could effectively trigger tumor antigen-specific cellular immunity, thus producing strong antitumor effect. At the same time, Shao et al. developed an DNA supramolecular hydrogel vaccine (DSHV) system which could elicit potent antitumor humoral immunity (Figure 6B) [89]. In their design, B cell epitopes of tumor cells and T cell epitopes from tetanus toxoid were fused into one single strand having branched seven-lysine residue at its N-terminal, and loaded into DNA hydrogel containing CpG ODNs via electrostatic interaction. After intra-tumoral injection, sustained release of antigen epitopes and CpG ODNs form the DNA hydrogel could induce high titers of antibodies against the tumor antigen, thus effectively inhibiting tumor growth.

DNA hydrogel containing CpG sequences has been demonstrated to be a valuable antigen delivery system. However, improving its sustained-release effect could further boost its potential as a delivery platform. In this regard, Ishii-Mizuno chose to incorporate chitosan, a cationic compound with good biocompatibility, into a hydrogel made up of negatively charged DNA strands [58]. This improved method made the hydrogel tougher and more stable, so it could provide a sustained release effect on the contents. In addition to electrostatic interactions, hydrophobic interactions are also a common mechanism for sustained release of DNA hydrogel inclusions. Umeki proposed to use the DNA hydrogel modified by hydrophobic compound cholesterol to realize the slow-release of model protein ovalbumin [94]. If this model protein could be extended to specific tumor antigens, it would greatly optimize current DNA hydrogels for tumor antigen delivery.

### 4.4. Photothermal Immunotherapy

As mentioned above, photothermal therapy is a cutting-edge cancer treatment method that uses materials with good photothermal conversion effects to kill tumor cells. Utilizing photothermal to trigger systematic immune reactions is a promising way of cancer treatments [95,96], and DNA hydrogel could act as a good bearer of photothermal immunotherapy. There is also a class of DNA hydrogels that carry both an immunostimulatory factor and a photothermal effect that kills tumor cells in a two-pronged way. Nishikawa et al. designed a composite immunostimulatory DNA hydrogel consisting of a hexapod-like structured DNA (hexapodna) containing a CpG sequence and gold nanoparticles [97]. Laser irradiation of the hydrogel released hexapodna, which could effectively stimulate the release of pro-inflammatory cytokines from immune cells. Laser irradiation also increased local temperature and mRNA expression of heat shock protein 70 in the tumor tissue, which enhanced immune responses. This hydrogel could significantly delay tumor growth and prolonged the survival of tumor-bearing mice. In addition, Oh et al. designed a DNA hydrogel loaded with CpG and c-di-GMP and coated with melanin for immunotherapy combined with photothermal therapy (Figure 6C) [98]. Melanin provided photoresponsiveness and inducing the calreticulin transport in cells after NIR exposure. This DNA hydrogel produces the photothermal killing of primary tumors and induces the maturation of dendritic cells (DCs) in lymph nodes, thus triggering a systemic immune response to inhibit the growth of distant tumors and prevent tumor recurrence.

## 5. Conclusions and Perspectives

In this review, we focused on the application of DNA hydrogels in local drug delivery and immunotherapy. We divided DNA hydrogels into pure DNA hydrogels and hybrid DNA hydrogels, and introduced their advantages, respectively. Then, the applications of DNA hydrogels in the field of local drug delivery, including antitumor drugs delivery, anti-inflammatory drugs delivery, anti-nerve damage drugs delivery and antibacterial drug delivery, were introduced in detail. Then, we also introduced the applications of DNA hydrogels in immunotherapy, including immune checkpoint inhibitor delivery, immune adjuvant delivery, tumor vaccine delivery and photothermal immunotherapy. DNA hydrogels are promising in controllable drug delivery and immunotherapy due to their unique properties of programmability, designable responsiveness, biocompatibility and biodegradability. These properties overcome the disadvantages of traditional drug delivery materials, such as non-biodegradability, uncontrollable mechanical properties and relatively high toxicity.

However, there are still limitations for the clinical applications of DNA hydrogels. (1) Chemically synthesized DNA as the raw materials of DNA hydrogels is relatively expensive, which limits the preparation of DNA hydrogels on a batch scale. Therefore, the cost should be minimized, for example by combining DNA with other naturally degradable backbones to prepare hydrogels to extend their application. (2) The pharmacokinetic and long-term biosafety of DNA hydrogels are not yet clear, limiting their use in clinical aspects. It is necessary to further clarify the absorption, distribution, metabolism and excretion of DNA hydrogels in vivo. At the same time, a systematic toxicity assessment must be carried out, especially for long-term toxicity studies. (3) The release characteristics of DNA hydrogels also need to be clarified and optimized. More precise release mechanisms need to be designed, such as drug delivery via dual-responsive hydrogels, etc., to make drug release systems more accurate and targeted. (4) The immune system in vivo is highly resistant to foreign DNA nanostructures, which may lead to the elimination of DNA hydrogels. Therefore, the immune response to exogenous nucleic acids should be minimized and harmful reactive residues should be removed to reduce unnecessary immune responses.

Overall, due to the unique advantages of DNA hydrogel, it would still be a promising material in biomedical engineering in the future. Moreover, it is expected that combining DNA hydrogels with immunomodulatory drugs would open new avenues for programmed immune modulation, which is attractive in the immunotherapy of many diseases.

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
