# Peer review of "Injectable DNA Hydrogel-Based Local Drug Delivery and Immunotherapy"

_gels, 2022, doi:10.3390/gels8070400_

Round 1
Reviewer 1 Report
The manuscript "Injectable DNA hydrogel-based local drug delivery and immunotherapy" summarizes the types of DNA hydrogels and the approaches for their formation to design and engineer systems for controllable drug delivery.
The paper is a comprehensive review in the area of drug delivery facilitated by DNA hydrogels. The authors were able to consider the most important types of DNA hydrogels. The list of references covers papers devoted to DNA hydrogels from the very beginning of the study of two stranding helical structures (in 1956) to recent achievements (in 2021-2022).
The submitted manuscript is well-prepared and needs just minor revision to be accepted for publication.
1. Figures 1 and 6 need to be restructured to provide a better resolution.
2. Conclusion should be enriched with an outlook devoted to challenges facing DNA hydrogels and future development directions.
3. Please double-check misprints and English language style.
Author Response
The paper is a comprehensive review in the area of drug delivery facilitated by DNA hydrogels. The authors were able to consider the most important types of DNA hydrogels. The list of references covers papers devoted to DNA hydrogels from the very beginning of the study of two stranding helical structures (in 1956) to recent achievements (in 2021-2022).The submitted manuscript is well-prepared and needs just minor revision to be accepted for publication.
- Figures 1 and 6 need to be restructured to provide a better resolution.
Reply: Thanks for your kindly and very helpful suggestions. We have restructured Figure 1 and increased the resolution of Figures 1 and 6 .
- Conclusion should be enriched with an outlook devoted to challenges facing DNA hydrogels and future development directions.
Reply: We have supplemented the challenges and future development directions of DNA hydrogels in the conclusion part.
- Please double-check misprints and English language style.
Reply: We have checked the main text again, and corrected some misprints and grammar problems.
Reviewer 2 Report
The manuscript entitled “Injectable DNA hydrogel-based local drug delivery and immunotherapy” by Wang et al. provides a comprehensive overview over recent advances of DNA hydrogels and their applications. The authors firstly focus on their classification and preparation methods and then, in different sections, research progress of injectable DNA hydrogels is reviewed. Challenges, drawbacks, limitations as well as future development are critically discussed. There exists an increasing activity in the field and such a comprehensive description of various approaches, preparation, and applications of the injectable DNA hydrogels is timely and will, most probably, cause interest of many researchers. The manuscript is well-structured with a proper balance between the different sections. It is well-written with a very few typos/grammar mistakes that can be easily corrected by a language editor. Practically, I do not have any serious suggestions and recommend accepting of the manuscript.
Author Response
The manuscript entitled “Injectable DNA hydrogel-based local drug delivery and immunotherapy” by Wang et al. provides a comprehensive overview over recent advances of DNA hydrogels and their applications. The authors firstly focus on their classification and preparation methods and then, in different sections, research progress of injectable DNA hydrogels is reviewed. Challenges, drawbacks, limitations as well as future development are critically discussed. There exists an increasing activity in the field and such a comprehensive description of various approaches, preparation, and applications of the injectable DNA hydrogels is timely and will, most probably, cause interest of many researchers. The manuscript is well-structured with a proper balance between the different sections. It is well-written with a very few typos/grammar mistakes that can be easily corrected by a language editor. Practically, I do not have any serious suggestions and recommend accepting of the manuscript.
Reply: Thanks very much for your helpful comments on our manuscript.
Reviewer 3 Report
The manuscript entitled “Injectable DNA hydrogel-based local drug delivery and immunotherapy” reviewed the preparation of DNA hydrogel for drug delivery systems, particularly immunotherapy. This paper focuses on the preparation strategy of DNA hydrogels and their application of local drug-releasing, and immune activation against cancer. The conclusions of this paper are reasonable, the advantages of DNA hydrogels are appealed in this paper compared to undegradable materials. The perspectives of this paper are also reasonable and informative for researchers in the field of material sciences. This review paper is suitable for the publication of Gels. I have a few relatively minor comments, as listed below.
1. Line 171: K+ should be written as K+. + should be a superscript.
2. Line 344: Staphylococcus aureus should be italic type.
3. Line 510: In situformation should be In situ formation.
4. Line 560: ochratoxin a should be ochratoxin A.
5. Line 565: 2d photonic should be 2D photonic.
Author Response
The manuscript entitled “Injectable DNA hydrogel-based local drug delivery and immunotherapy” reviewed the preparation of DNA hydrogel for drug delivery systems, particularly immunotherapy. This paper focuses on the preparation strategy of DNA hydrogels and their application of local drug-releasing, and immune activation against cancer. The conclusions of this paper are reasonable, the advantages of DNA hydrogels are appealed in this paper compared to undegradable materials. The perspectives of this paper are also reasonable and informative for researchers in the field of material sciences. This review paper is suitable for the publication of Gels. I have a few relatively minor comments, as listed below.
- Line 171: K+ should be written as K+. + should be a superscript.
Reply: We have corrected this typo.
- Line 344: Staphylococcus aureus should be italic type.
Reply: We have corrected this typo.
- Line 510: In situformation should be In situ formation.
Reply: We have corrected this typo.
- Line 560: ochratoxin a should be ochratoxin A.
Reply: We have corrected this typo.
- Line 565: 2d photonic should be 2D photonic.
Reply: We have corrected this typo.